# Impact of COVID-19 Pandemic on Women’s Health and Obstetric Outcomes after Assisted Reproduction: A Survey from an Italian Fertility Center

**DOI:** 10.3390/jpm13030563

**Published:** 2023-03-21

**Authors:** Michela Cirillo, Valentina Basile, Letizia Mazzoli, Maria Elisabetta Coccia, Cinzia Fatini

**Affiliations:** 1Department of Experimental and Clinical Medicine, University of Florence, 50134 Florence, Italy; michela.cirillo@unifi.it; 2Centre for Assisted Reproductive Technology, Division of Obstetrics and Gynaecology, Careggi University Hospital, 50134 Florence, Italy; valentina.basile@unifi.it (V.B.);; 3Department of Clinical and Experimental Biomedical Sciences, University of Florence, 50134 Florence, Italy

**Keywords:** assisted reproductive techniques, COVID-19, pregnancy, Mediterranean diet, lifestyle, preventive medicine, emotional state, women’s health

## Abstract

Background: the restrictive measures that were adopted during three waves of the COVID-19 pandemic had an impact on both the emotional state and lifestyle of the general population. We evaluated the impact of COVID-19 pandemic on lifestyles and emotional states of women planning assisted reproductive technology (ART), and whether these changes affected ART outcomes. Methods: quantitative research, using a web-based survey, was performed on 289 Caucasian women. Results: In preconception, we observed higher percentage of women with positive obstetric outcomes who reduced body weight (52.4% vs. 27.2%, *p* = 0.09). Over 60% of women with positive outcomes practiced physical activity vs. 47% of women with negative outcomes (*p* = 0.03), as well as having better quality of sleep (45% vs. 35%), and a more solid relationships with their partners (65.1% vs. 51.7%, *p* = 0.03). Women who increased their intake of whole grains, fruits, vegetables, and legumes (*p* < 0.05), according to the Mediterranean diet, showed positive outcomes. We observed that participants who experienced “very much” or “extreme” anxiety, sadness, and fear (*p* < 0.05) during pandemic were clearly more numerous in the group with negative pregnancy outcomes. Conclusions: healthy lifestyle together with a positive emotional state in preconception can positively influence the obstetric outcomes after ART.

## 1. Introduction

The COVID-19 pandemic had a significant impact on the field of reproductive medicine: in Italy on 18 March 2020, the Group of Special Interest on Sterility (GISS) of the Italian Society of Gynecology and Obstetrics (ISGO) and its federates declared the suspension of all the Assisted Reproductive Technology (ART) treatments to prevent the moving of people and their access to clinical facilities, in order to contain the spread of the virus [1]. Exceptions were made for ongoing stimulation cycles and fertility cryopreservation procedures in oncological patients [2].

The Italian infertile population is the oldest among Western countries: in about a third of cases, the female partner is over the age of 40, and time is fundamental to achieve a good outcome of ART treatments [3]. Moreover, in Italy, every month of ART inactivity causes a failure to the performance of about 8000 treatments with a potential falling of the birth rate of about 1500 children. Therefore, from May 2020 onwards, when the general lockdown was already left behind, all reproductive medicine activities resumed [4].

Infertility is a distressing condition that induce emotional, financial, and social strain [5]. The COVID-19 pandemic intensifies this stressful status in infertile women attempting ART, due to its impact on women’s life (isolation, economic shutdowns, unemployment, and fear of contamination) [6].

However, the restrictive measures that were adopted during the first, the second and the third wave of the pandemic had an impact not only on the emotional state (fear, anxiety, and distress), but also on the dietary habits and lifestyle of the general population. These two aspects, associated with systemic inflammation and endothelial dysfunction, play a major role in the development and in the progression of cardiovascular diseases [7,8].

As a matter of fact, people having a positive psychological attitude tend also to have healthy habits, such as a healthy diet, an active lifestyle, and a low incidence of sleep-related disorders. In addition, the Mediterranean diet has positive effects on health, as it reduces the risk of developing cardiovascular diseases, the incidence of developing malignant tumors, neurodegenerative diseases, and diabetes [9]. On the other hand, during the first wave of the pandemic, women waiting to undergo ARTs have shown changes in weight, an increase of BMI, a high consumption of sweets, cheese, meat, and snacks [10]. Besides several gynecological and systemic diseases affecting women’s fertility, lifestyle factors and environmental conditions, such as stress induced by COVID-19 pandemic, contribute to interfere with reproductive health in both women and men. The literature shows that an unbalanced and unhealthy diet can lead to adverse ART outcomes, while physical activity and healthy eating can lead to improvements on menstrual cycles and female fertility [11]. Specifically, nutritional factors may affect not only oocyte maturation, but also quality of embryos and implantation [12].

This study aimed to evaluate the impact of COVID-19 pandemic restrictive measures on lifestyle and emotional state of women who have undergone ART from May 2020 to February 2021, and whether changes related to pandemic period had affected or not ARTs’ outcomes.

## 2. Materials and Methods

We performed a study in a sample of Caucasian women, referred to the Internal Medicine Clinic at the Assisted Reproductive Technology Centre, using a web-based survey. The survey was conducted in Italian, according to the CHERRIES (Checklist for Reporting Results of Internet E-Surveys) Statement [13]. The Italian version of the questionnaire was created online by using Google Forms. We sent the questionnaire to 480 women by email using a validated account of the University of Florence. The survey was addressed from 3 May to 31 July 2021. We included women aged between 18–49 years, planning homologous or heterologous infertility treatments at ART Centre from May 2020 to February 2021. Non-Caucasian women were excluded. An information sheet was set as the first page of the web survey, and participants had the opportunity to give informed consent, according to Ethical principles of the Declaration of Helsinki, before accessing the survey. All the potential participants were fully informed about the study, the extent of privacy, anonymity and confidentiality, the voluntary nature of participating, and the lack of negative consequences in case of decline.

The Local Ethics Commitee (Azienda Ospedaliero-Universitaria Careggi) approved the study (Prot. 19901_OSS). The 46 self-administered questions were designed to assess the impact of COVID-19 pandemic on lifestyle habits as well as psychological behavior and obstetric outcomes of the ART procedure. Basic demographic data were recorded: age, level of education, and region of domicile. Socio-economic status comprised types of job before and during pandemic. Clinical data included: types of procedures (homologous or heterologous), height, weight gain during the pandemic (Body Mass Index, BMI, calculated by dividing weight in kilograms by the square of height in meters. According to the World Health Organization criteria, overweight was defined as BMI values ≥ 25 kg/m^2^), and presence of chronic diseases. Nutritional habits, physical activity, smoking habit, quality, and quantity of sleep during the pandemic were evaluated (multiple choice). The emotional state, including anxiety, sadness, and fear during the pandemic, using a verbal rating scale (not at all, slightly, moderately, very much, extremely) was assessed.

### Statistical Analysis

Statistical analysis was performed by using the SPSS (Statistical Package for Social Sciences, Chicago, IL, USA) software for Windows (Version 28.0). Continuous variables were expressed as mean (±SD). The categorical variables were expressed as frequencies and percentages. Chi-square test was used to test for proportions. The continuous variables were analyzed by using a parametric test (*t*-Student test). A *p*-value < 0.05 was considered significant.

## 3. Results

The questionnaire was sent to 480 women and was completed by 289 responders. Demographic, socio-economic, and clinical characteristics of the study population are reported in Table 1. The mean age was 39.4 (±4.7 years) and 118 responders (40.8%) were more than 40 years old. More than 70% of women came from Central Italian regions and more than 50% of women had graduated (graduation and post-graduation).

In comparison to our previous study performed during the first wave of pandemic [10], the percentage of women unemployed was lower (43.6% vs. 15.2%), as expected.

In Table 2, we report lifestyle changes during the COVID-19 pandemic. Specifically, we observed that about 18% of women reduced body weight and about 24% quit smoking. Indeed, about 54% of women practiced regularly physical activity (1–2 times a week in 29.8%, 3–4 times a week in 18.3% and more than 4 times a week in 5.5%), a higher percentage in comparison to that observed in our previous study during the first wave of the pandemic [10].

We investigated quality and quantity of sleep habits during the pandemic and about 26% of women reported feeling tired when waking up, about 28% of them had difficulties falling asleep or reported fragmentated sleep, while about 6% of the patients needed to take medications or supplements to treat sleep disorders. In our previous study, performed during the first wave of the pandemic, the percentage of women who reported fragmentated sleep or took medications or supplements to treat sleep disorders, was about twice as high (60% vs. 34%) [10].

### 3.1. COVID-19 Pandemic and Diet

We evaluated the habitual food intake of women. Their usual water intake was not optimal: only 9.7% of women drank more than two liters of water a day. About 65% of the participants did not usually drink other beverages such as sugar drinks (beverage with added sugar or other sweeteners such as high fructose corn syrup, sucrose, fruit juice concentrates), and this habit did not change during the pandemic.

Most of the women did not change their intake of alcohol and coffee during the pandemic: as concerns alcohol consumption, we observed that the 51% were non-drinkers, the 23% drink during the weekend and 21% of participants reduced alcohol consumption before ART.

We also investigated the consumption of unplanned snacks during the pandemic and before ART. We focused on women’s motivation for snack consumption and we observed that most of the participants (37.7%) ate snacks because of hunger, and more than 20% ate snacks because of boredom or of anxiety. The answers most frequently selected as a response to the question of which type of food was eaten as unplanned snacks was fresh fruit in 37.7% of women, yogurt in 30.1%, pastries, and savory snacks (pizza, chips, salted peanuts) in 28.2% of women.

Moreover, we evaluated the adherence to the Mediterranean diet pattern, and we observed that 34% of the women ate whole grain; in addition, the 13.8% of them declared that they started eating whole grain before ART procedure. Concerning the consumption of vegetables and fruits, we observed that a large percentage of women did not eat vegetables (2.4%) and fruits (8.3%) or did not eat them in adequate quantities (41.1% vegetables one portion a day; 50.2% fruit 1one portion a day). In relation to legume consumption, about 60% ate them once a week and 5.5% started or increased legume intake before ART.

Moreover, about 50% of women ate fish once a week, and 9% of women increased fish intake (two-to-three times a week) before ART procedure. On the other hand, regarding red and/or processed meat consumption, we observed that about 61% of participants ate them with a frequency of one-to-two times a week, 18% reduced consumption to less than two times a week but 8.6% of women consumed them more than two times a week during COVID-19 pandemic and before ART.

### 3.2. COVID-19 Pandemic and Emotional State

We investigated emotional state during COVID-19 pandemic and before ART procedure. We analyzed levels of anxiety, sadness, and fear in the population, using a verbal scale, and the reasons that determined emotion.

We found that about 34% of participants experienced moderate anxiety during the pandemic: 38.4% of the population felt anxious because of the restrictions following the COVID-19 pandemic, 20.4% felt anxious due to the future ART procedure, and 12.8% for the ART outcome. As concerns sadness, we found similar percentages among the options “not at all”, “slightly”, “moderately” (respectively, 26%, 28%, 23.5%), whereas 19.7% selected the option “very much”. The most frequent cause of sadness was economic condition, and in 17.6% lack of parenthood. Regarding fear, we found that a high percentage (45%) of women selected the option “slightly” and 27% “moderately”. Of the sample, 45.3% said they felt fear for “failing to carry out the ART procedure because of pandemic restrictions”.

We analyzed the emotional reactions to the feelings felt, allowing the participants to select more than one possible option. Interestingly, we noted that the most-selected response was “I sought comfort in the partner” (70.6%), followed by “I sought comfort in family members” (28.7%). When asked if the partner had been supportive during the pandemic and the ART procedure, 94.8% of participants answered “yes”. In addition, we investigated the reasons that motivated them to continue the ART plan despite the difficulties resulting from COVID-19 pandemic: the majority of participants (96.9%) said they were driven by a strong desire for motherhood.

### 3.3. COVID-19 Pandemic and Obstetric Outcomes

Obstetric and pregnancy outcomes of all women are reported in Figure 1. We observed that 143 (49.5%) of women had implantation failure, 24 (8.3%) had pregnancy and delivery, 59 (20.4%) had ongoing pregnancy, 26 (9%) had miscarriage, 34 (11.8%) stopped treatment for several gynecological reasons, and 3 (1%) stopped treatment for COVID-19 infection. We analyzed the arising emotions from the failure of the ART procedure and allowed participants to select as many options as possible; finding that sadness (26.3%) and disappointment (18%) were the emotions that they felt most.

We compared the lifestyle and the emotional state in relation to the outcomes of the ART procedure, to understand the influence that these variables may have had on the ART outcomes (Figure 2, Figure 3 and Figure 4).

In Figure 2, we observed a higher percentage of women with overweight/obesity who reduced their body weight before ART procedure by enhancing BMI in the group of women with positive outcomes (52.4% vs. 27.2%, *p* = 0.09). In addition, in the preconception period, over 60% of women with positive outcomes practiced physical activity compared to the 47% observed in the group of women with negative outcomes (*p* = 0.03). Concerning smoking habits, women who obtained positive outcomes were more virtuous, and had a better quality of sleep (45% vs. 35%), as well as having more solid relationships with their partners (65.1% vs. 51.7%, *p* = 0.03).

Concerning eating habits (Figure 3), women who increased their adherence to the Mediterranean pattern in their preconception period had more positive outcomes. In particular, in women with positive outcomes we observed an increase in the intake of whole grains (*p* = 0.02), fruits (*p* = 0.04), vegetables (*p* = 0.01), and legumes (*p* = 0.05), and a reduction in the consumption of sweets and sugar drinks, even if not significant. Before ART the alcohol consumption was low, according to the Mediterranean diet, in women with positive obstetric outcomes in comparison to women with negative outcomes (*p* < 0.0001).

Finally, regarding emotional state (Figure 4), we observed that participants who experienced “very much” or “extreme” anxiety (*p* = 0.001), sadness (*p* = 0.002), and fear were clearly more numerous in group with negative pregnancy outcomes.

## 4. Discussion

The COVID-19 pandemic and the restrictions imposed on the population had important effects on people’s lifestyle and emotional state, especially on women planning ART [10].

To date, evidence confirms that implementing good practices in lifestyle significantly reduces fertility problems, improves pregnancy outcomes, and generally provides for a good state of health throughout life [14,15]. On the other hand, the COVID-19 pandemic affected personal freedom, led to uncertainty about ART procedures, and also had a negative impact on the psychological sphere and lifestyle [10,16].

To the best of our knowledge, no other studies have considered the impact of the COVID-19 pandemic—which we are still going through—on lifestyle and emotional state in relation to ART outcomes.

Results coming from this survey showed that, in preconception, improving lifestyle in terms of better adherence to the Mediterranean diet pattern, adequate BMI, increased physical activity, and quitting smoking were associated with a higher probability of positive obstetric outcomes.

It is well-known that female overweight/obesity has a significant harmful effect on live birth rate following ART, possibly due to impaired ovarian folliculogenesis, oocyte quality, embryonic development, and uterine environment [17]. Moreover, obese pregnant women are at established increased risk for maternal, perinatal, and fetal complications [18]. In our study, we observed a higher percentage of women with overweight/obesity who reduced body weight before ART procedure by enhancing BMI in the group of women with positive outcomes, thus permitting to hypothesize that preconceptional maternal weight loss might be benefit before ART and during pregnancy. The literature focusing on weight loss suggested that a weight reduction of 5–10% correlate with dietary and lifestyle improvements; moreover, weight loss is often sufficient to improve the chance of pregnancy and metabolic parameters [19].

Indeed, it is well known that adherence to the Mediterranean diet significantly reduces the risk of developing cardiovascular diseases, neurodegenerative diseases, diabetes, cancer development, and overall mortality [20]. Moreover, scientific evidence points out that the Mediterranean diet, rich in vegetables, fruits, whole grains, legumes, extra virgin olive oil, fish, and a reduced intake of red and/or processed meat determines a greater probability of success for ART, as well as preserving and improving fertility [21,22].

Most couples who refer to the ART Centre searching for pregnancy are extremely motivated to take any necessary measures to maximize the procedure chances of success. Our study showed that a good percentage of participants improved their nutrition habits during the pandemic in accordance with the Mediterranean diet pattern.

In our study, which covered a period of about a year when restrictions were weaker if compared to the first phases of the pandemic, we observed that virtuous behaviors during preconception were more evident in women who had a positive outcome after ART.

On the other hand, it is worth noting that 60% of women with negative outcomes consumed unplanned snacks (especially pastries and savory snacks), of whom about 27% did so because of boredom and anxiety. This emotional eating may contribute to excess energy intake and weight gain, as also observed in our previous study [10]. In keeping with this, recent scientific findings suggest that food consumption is an important factor that affects the occurrence of mental illness [23].

It is noteworthy that a diet rich in ultra-processed foods (UPFs), saturated fat, sugar, salt, strongly flavored ingredients, and chemical additives is related to an increase in neuroinflammation. UPFs’ formulation, presentation, and marketing often promote overconsumption, and the evidence so far shows that UPFs’ consumption is associated with unhealthy dietary nutrient profiles. In contrast, the consumption of complex carbohydrates, such as vegetables, fruits, and fiber, is recommended because they are sources of vitamins and polyphenols, and are metabolized into short-chain fatty acids, which are important anti-inflammatory agents [24].

Meanwhile, we must emphasize that following or switching to a Mediterranean diet pattern does not only imply making better food choices, but it means adopting a real and healthier lifestyle, characterized by regular physical activity, adequate rest, and conviviality, too; accordingly, we observed that in the group of women with positive outcomes, about 60% continued to exercise regularly compared to 47.6% of patients who had later a negative outcome. Several studies evaluated the effect of exercise on ART outcomes with conflicting results [25]. Some data indicate that physical activity may have beneficial effects on some reproductive health outcomes in young adult women, even if the type, intensity, frequency, and the role of physical activity independent of weight loss remain unclear [26]. On the other hand, we well know that sedentary behavior and sitting time (while watching TV) are associated with increased risk of cardiovascular disease and with several cardiometabolic and mental effects [8].

It is known that sleeping adequately helps to regulate appetite and to reduce cardiovascular risk [27]; in our study we found out that about 60% of the participants who reported having difficulty falling asleep or felt tired when waking up needed to take medication/supplements to treat sleep disorders. Despite this, we have not observed significant differences in sleep quality between the two groups in relation to obstetric outcome.

Not surprisingly, our data regarding the emotional status of women in relation to clinical outcomes showed that anxiety and sadness during the pandemic were associated with a higher frequency of negative outcome after ART. Purewal et al., evidenced that depression and anxiety during ART treatment are associated with poor ART outcome, nevertheless no evidence about changes in the levels of anxiety and depression from baseline to ART procedures are associated with ART outcome [28].

All of that could have a negative impact, as the literature data have confirmed that negative emotional factors, such as anxiety, sadness and stress, are considered predisposing factors potentially increasing the risk of cardiovascular diseases, also through potential influences on the lifestyle [29].

A recent survey capturing emotional reactions of people during lockdown, and before fertility clinics announced re-opening, reported more negative than positive emotions, in particular stress, worry, frustration, and anger [30]. Negative emotional state, in particular anxiety and sadness, was possibly associated with poorer ART outcome. These findings could help to identify women needing tailored psychological support during the different stages of infertility treatments [31,32].

Therefore, our findings suggest the necessity to implement adequate measures such as meetings and tailored information for the couples, with the aim to raise awareness about the importance of health in the preconception period, especially regarding lifestyle and nutrition. Moreover, all couples who are at risk to experience negative emotional states should be readily identified and referred to appropriate counselling to improve the chances of ART success.

The limitations of this research include the lack of validated questionnaire; nevertheless, the survey was able to focus on specific group of women planning ART. Moreover, another limitation is represented by a self-reported questionnaire, which may be associated with the possibility of bias (social desirability), as well as the lack of inclusion of men’s behaviors and perspectives.

## 5. Conclusions

In conclusion, a healthy lifestyle, characterized by regular physical activity, a normal BMI, a stable socioeconomic condition, a good quality of sleep, and a good diet in line with the Mediterranean diet pattern, together with a positive emotional state and an established relationship, can positively influence the obstetric ART outcomes.

The creation of a path dedicated to women’s preconception health in ART requires a multidisciplinary approach in which professional figures such as gynecologists, midwives, dietitians, phycologists, and general practitioners cooperate for health promotion, during ART procedure and pregnancy, but also for women’s health later in life.

## Figures and Tables

**Figure 1 jpm-13-00563-f001:**
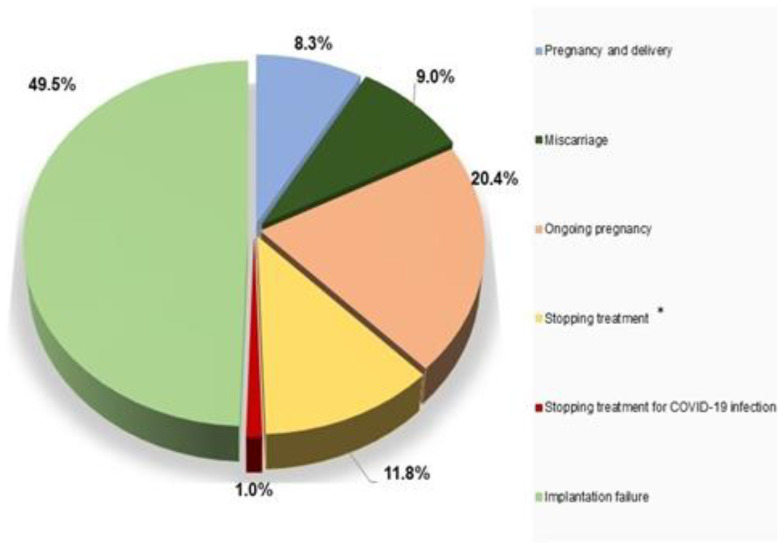
Percentages of obstetric and pregnancy outcomes after ART in women investigated. * Stopping treatment indicates that ART treatment was stopped for one of these reasons: under stimulation, over stimulation, no or low fertilization, delayed or abnormal embryo development.

**Figure 2 jpm-13-00563-f002:**
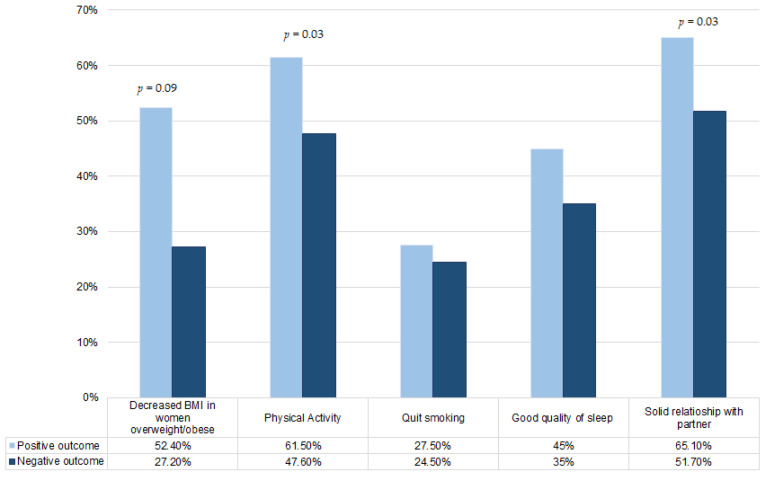
Distribution of variables (percentage of overweight/obese women who improved BMI, percentage of women who practiced physical activity, percentage of women who quit smoking, percentage of women who had good quality of sleep, percentage of women who had a solid relationship with partner) in women with positive ART outcomes vs. women with negative ART outcomes.

**Figure 3 jpm-13-00563-f003:**
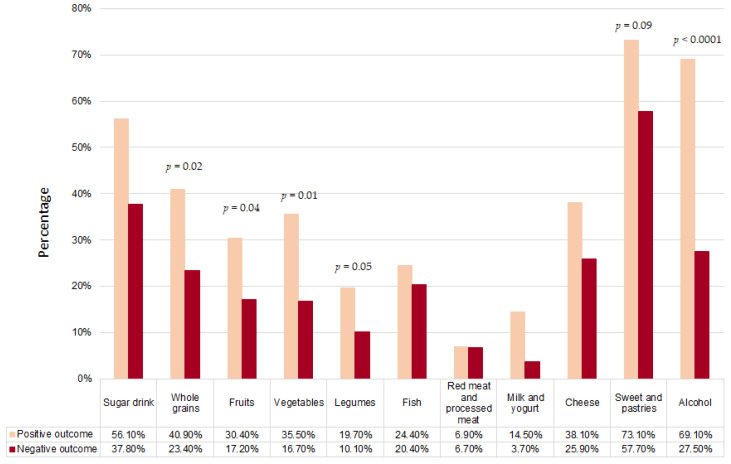
Distribution of women who increased their adherence to Mediterranean diet for each group of food according to ART outcomes (positive outcome vs. negative outcome).

**Figure 4 jpm-13-00563-f004:**
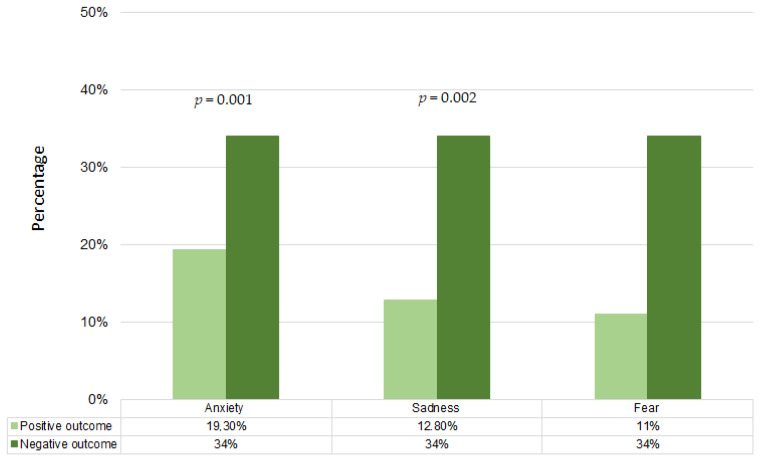
Distribution of women experienced anxiety, sadness, and fear, during COVID-19 pandemic according to ART outcomes (positive outcome vs. negative outcome).

**Table 1 jpm-13-00563-t001:** Demographic, socio-economic, and clinical characteristics of study population.

Variables	n = 289
Age, yrs	39.4 ± 4.7
Age > 40 yrs, n (%)	118 (40.8)
BMI ≥ 25 kg/m^2^, n (%)	52 (17.9)
Smoking habit, n (%)	26 (9.0)
Homologous ART, n (%)	148 (51.2)
Eterologous ART, n (%)	141 (48.8)
First-time ART, n (%)	128 (44.3)
Geographical distribution, n (%)	
Northern Italian Regions	58 (20.0)
Central Italian Regions	215 (74.4)
Southern and Islands Italian Regions	16 (5.5)
Level of Education, n (%)	
Middle School	24 (8.3)
High School	111 (38.4)
Graduation	106 (36.6)
Post-Graduation	48 (16.6)
Work during COVID-19 pandemic, n (%)	
Active work outside	170 (58.8)
Smart working	75 (25.9)
Unemployed	44 (15.2)

Values are reported as mean ± SD or n, (%). ART (Assisted Reproductive Technology).

**Table 2 jpm-13-00563-t002:** Lifestyle changes during COVID-19 pandemic.

	n = 289
**During last year and before ART, did you change your body weight?**	
No changes	84 (29.1)
Increased (1–5 kg)	114 (39.5)
Increased (>5 kg)	40 (13.8)
Reduced (1–4 kg)	35 (12.1)
Reduced (>4 kg)	16 (5.5)
**During last year and before ART, did you smoke?**	
Never	195 (67.4)
Quit	69 (23.8)
Reduced	23 (7.9)
No changes	2 (1.4)
**During last year and before ART, did you regularly practice physical activity?**	
No, I didn’t do any physical activity even before	82 (28.4)
No, I have interrupted habitual physical activity	52 (18)
Yes, 1–2/week	86 (29.8)
Yes, 3–4/week	53 (18.3)
Yes, >4/week	16 (5.5)

Values are reported as n, (%). ART (Assisted Reproductive Technology).

## Data Availability

Not applicable.

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
