# Peer review of "Impact of COVID-19 Pandemic on Women’s Health and Obstetric Outcomes after Assisted Reproduction: A Survey from an Italian Fertility Center"

_jpm, 2023, doi:10.3390/jpm13030563_

Round 1
Reviewer 1 Report
The paper by Cirillo et al. titled “Impact of COVID-19 Pandemic on women’s health and obstet- 2 ric outcomes after Assisted Reproduction: A survey from an Italian Fertility Center” evaluates the impact of COVID-19 pandemic on lifestyle and emotional state of women planning assisted reproductive technology (ART) and its outcome.
Quantitative research involves the utilization and analysis of numerical data using specific statistical techniques to answer questions like who, how much, what, where, when, how many, and how.
It seems that this study does not answer all these questions.
Introduction
Line 32-33 – modify the acronym accordingly - Italian Society of Gynecology and Obstetrics (SIGO) – ISGO
Material and Methods
You stated that “We included women aged between 18–49, 69 planning homologous or heterologous infertility treatments at ART Centre from May 2020 70 to February 2021.” – How do you explain the inclusion of very young women in the study? It would have been very useful to understand this. Did you ask them about Anti-Müllerian hormone used as a biomarker of a woman’s ovarian reserve, along with levels of follicle-stimulating hormone and antral follicular count? Studies are showing that the largest cohort of women receiving ART treatments were those between 35 and 40 years. What type of ATR did they use, where was the biggest changes found?
How many did not succeed in ART treatments, how many are pregnant during the survey? How many experienced miscarriages?
Results
This section is hard to follow in my opinion. It is not clear in Figure 1, due to the colors used in the pie graphics to distinguish between implantation failure and ongoing pregnancy. The figure legend should be self-explanatory, which is not. *Stopping treatment for under stimulation, over stimulation, no or low fertilization, delayed or abnormal embryo development. – this is completely unclear. The asterisk refers to what?
Figure 2-4. You should write on the ordinate what is representing. Also, you should describe in the legend the outcomes.
Discussion
Are there any studies involving the same parameters before the pandemic? Compare your date with the values found in the literature involving diet, BMI and stress and their impact on ART procedures.
Extensive English editing is needed, some paragraph are difficult to follow.
Author Response
Thank for your attention and care in reading and reviewing our paper.
Introduction
-We modified the acronym SIGO in ISGO, according to your request.
Materials and Method
-Many thanks for your question.
Italian laws grant the access to ART procedures to infertile couples with people of age 18 or more, of different sex, married or cohabitants, in their potentially fertile age, only.
University Hospital of Careggi ART Center is a third level center, with thousands of cycles performed every year, with people coming from all the Italian regions, including very young couples. Accordingly, the age range considered for the survey was the one permitted by law. Although it is rare to visit very young women, it is still possible, since some of them may present Premature Ovarian Failure. However, the median age of the sample was 39 years (with a 26-49 range).
-In the results we reported the number and percentage of miscarriages, ongoing pregnancy, and ART failure, as required (page 7 lines 205-209).
-We improved the results, trying to make it clear.
-We modified the colours used in figure 1 and we clarified the legend, in particular the sentence with asterisk.
-In figures 2-4, we indicated on the ordinate the meaning, as required. Moreover, we clearly described the outcomes in the legends.
Discussion
-According to your suggestion, we compared our findings with data from literature.
We added sentences and references (page 12 lines 274-283, page 13 lines 314-315, 327-330).
Extensive English editing is provided.
Reviewer 2 Report
The manuscript is quite well written. It contains all the necessary and relevant data. The statistics are in accordance with the analyzed data. The text is easy to read and understand. I consider the manuscript acceptable for publication.
Author Response
Thank for your attention and care in reading and reviewing our paper.